# Genome-Wide Identification of C2H2 ZFPs and Functional Analysis of *BRZAT12* under Low-Temperature Stress in Winter Rapeseed (*Brassica rapa*)

**DOI:** 10.3390/ijms232012218

**Published:** 2022-10-13

**Authors:** Li Ma, Jia Xu, Xiaolei Tao, Junyan Wu, Wangtian Wang, Yuanyuan Pu, Gang Yang, Yan Fang, Lijun Liu, Xuecai Li, Wancang Sun

**Affiliations:** 1State Key Laboratory of Aridland Crop Science, Gansu Agricultural University, Lanzhou 730070, China; 2College of Agronomy, Gansu Agricultural University, Lanzhou 730070, China

**Keywords:** winter *B. rapa*, *BrZAT12*, C2H2 zinc-finger protein, cold resistance, abiotic stress

## Abstract

Zinc-finger protein (ZFP) transcription factors are among the largest families of transcription factors in plants. They participate in various biological processes such as apoptosis, autophagy, and stemness maintenance and play important roles in regulating plant growth and development and the response to stress. To elucidate the functions of ZFP genes in the low-temperature response of winter (*Brassica rapa* L.) *B. rapa*, this study identified 141 members of the C2H2 ZFP gene family from *B. rapa*, which are heterogeneously distributed on 10 chromosomes and have multiple *cis*-acting elements related to hormone regulation and abiotic stress of adversity. Most of the genes in this family contain only one CDS, and genes distributed in the same evolutionary branch share mostly the same motifs and are highly conserved in the evolution of cruciferous species. The genes were significantly upregulated in the roots and growth cones of ‘Longyou-7’, indicating that they play a role in the stress-response process of winter *B. rapa*. The expression level of the Bra002528 gene was higher in the strongly cold-resistant varieties than in the weakly cold-resistant varieties after low-temperature stress. The survival rate and *BrZAT12* gene expression of trans-*BrZAT12 Arabidopsis thaliana* (*Arabidopsis*) were significantly higher than those of the wild-type plants at low temperature, and the enzyme activities in vivo were higher than those of the wild-type plants, indicating that the *BrZAT12* gene could improve the cold resistance of winter *B. rapa*. *BrZAT12* expression and superoxide dismutase and ascorbate peroxidase enzyme activities were upregulated in winter *B. rapa* after exogenous ABA treatment. *BrZAT12* expression and enzyme activities decreased after the PD98059 treatment, and *BrZAT12* expression and enzyme activities were higher than in the PD98059 treatment but lower than in the control after both treatments together. It is speculated that BrZAT12 plays a role in the ABA signaling process in which MAPKK is involved. This study provides a theoretical basis for the resolution of cold-resistance mechanisms in strong winter *B. rapa*.

## 1. Introduction

Winter rapeseed (*B. rapa*) is one of the most important oil crops in the arid and cold regions of northern China. At present, the northward migration of winter rapeseed has been successfully carried out in the north, and the cultivation area of winter rapeseed has gradually changed because of its strong cold resistance and its ability to winter smoothly under the severe cold environment in the north [1,2]. The fact that winter rapeseed can successfully winter under low temperatures and shows good cold resistance indicates that it is rich in cold-resistance genes and is thus an important material for studying cold resistance.

During plant growth and development, abiotic stresses such as low temperature, high salinity, and drought affect plant growth and crop yield [3,4]. Plants have evolved protective mechanisms to cope with stresses. These involve morphological, physiological, and molecular adaptations over a long evolutionary period [5,6]. Transcription factors, also known as *trans*-acting factors, are an important regulator of the environment in plant cells. They can bind with promoters to activate gene transcription, participate in plant growth and development regulation, participate in signal transduction, and regulate gene expression under stress [7,8]. Therefore, transcription factors related to plant-stress responses have been the focus of much research. The transcription factors that have been identified in plants that play a role when plants encounter adverse environments include NAC, ZFP, bHLH, bZIP, and WRKY [9,10].

ZFPs have been found to exist in a variety of plants and have various biological functions, such as regulating gene expression, DNA recognition, and participating in plant growth and development. A large number of ZFPs have been discovered, among the most popular ones are C2H2, which mostly relate to the stress response and plant growth and development regulation [11,12]. ZFPs are one of the most widely studied transcription factor families. According to their structure and characteristics, ZFPs can be divided into several subfamilies, including C2H2, CCCH, C3HC4, C2HC, C2HC5, C4, C4HC3, C6, and C8, where C stands for Cys (cysteine) and H for His (histidine) [3,11]. Different types of ZFPs have different spatial configurations, resulting in different functions of the proteins. Among them, the C2H2 ZFPs have the most eukaryotic members. Each ZFP contains two cysteine (Cys2) and two histidine (His2) residues, which bind stably to Zn^2+^ and specifically to a domain in the gene promoter regulated by the C2H2 ZFPs [11,13,14].

According to reports, members of the C2H2 ZFP gene family have been identified in a variety of plants, such as *Arabidopsis*, potato, sorghum, tomato, and poplar, and it has been indicated that C2H2 ZFPs in plants play an important regulatory role in abiotic stress responses [15,16,17,18,19]. The constitutive expression of *ZAT12* in *Arabidopsis* caused a small but reproducible increase in freezing tolerance, suggesting a role for the ZAT12 regulator in cold acclimation. In addition, ZAT12 downregulated *CBF* gene expression, suggesting a role for ZAT12 in a negative regulatory loop that inhibits the expression of the CBF cold-response pathway [20]. It was shown that the *BcZAT12*-transformed tomato lines ZT1, ZT2, and ZT6 were better adapted to drought stress by accumulating the osmotic protein proline and increasing the antioxidant response triggered by the *ZAT12* gene. Thus, the *ZAT12*-gene-transformed tomato *cv*. H-86 lines will prove useful for producing high-yield tomato crops in areas subjected to severe drought stress [21]. The overexpression of soybean *GmZF1* significantly improved the proline and soluble-sugar content and reduced the malondialdehyde (MDA) content in transgenic lines exposed to cold stress, suggesting that transgenic *Arabidopsis* carrying the *GmZF1* gene has an adaptive mechanism to cold stress. The overexpression of *GmZF1* also increased the expression of the cold-regulated *cor6.6* gene, probably by recognizing the protein–DNA binding site, indicating that *GmZF1* from soybean can improve cold-stress tolerance in *Arabidopsis* by regulating the expression of cold-regulated genes in transgenic *Arabidopsis* [22]. Based on stress-induction analysis, mutation analysis, complementation analysis, and ectopic-expression analysis, many C2H2 ZFPs involved in abiotic stress signaling pathways were identified [11]. Phytohormones are responsible for resistance to abiotic stresses and are involved in the process of the response to various stresses through C2H2 ZFPs, especially ABA [23,24].

In this study, we analyzed the gene chromosome distribution, gene evolution, motifs, *cis*-acting elements, and covariance of the C2H2 ZFP gene family in winter rapeseed (*B. rapa*) based on genomic and transcriptomic data and investigated the expression pattern of C2H2 ZFP genes under abiotic-adversity stress. The *BrZAT12* gene was identified in response to cold stress, and the gene was functionally validated. This study provides a theoretical basis for the analysis of the mechanism by which the C2H2 ZFP regulates cold resistance in winter rapeseed as well as for the cultivation of strong cold-resistant quality germplasm.

## 2. Results

### 2.1. Chromosome Distribution of the C2H2 ZFP Gene Family

The C2H2 ZFP gene family in *B. rapa* was identified by TBtools. A total of 141 members of the C2H2 ZFP gene family were identified. According to the visual analysis results of the chromosome distribution obtained by TBtools software (Figure 1, Appendix A), it was found that the members of the gene family were unevenly distributed on 10 chromosomes. Among them, most were distributed on chromosome A03, containing 24 members of the gene family, followed by A02 (18), A09 (18), A06 (17), A07 (15), A01 (12), A10 (12), A04 (10), and A05 (9), and the least were distributed on chromosome A08, including only six gene-family members.

### 2.2. Phylogenetic Analysis of the C2H2 ZFP Gene Family

Furthermore, the phylogeny of the C2H2 family members of *B. rapa* was studied. The phylogenetic relationships showed that the C2H2 ZFP family of *B. rapa*, *B. napus*, and *Arabidopsis* could be divided into 16 subfamilies (I–XVI) (Figure 2). The C2H2 ZFP family of *B. rapa* could be divided into 12 subfamilies (I–XII) (Figure 3A). Among them, subfamily IV contained, at most, 78 genes and the subfamily XI contained at least four genes. The genetic distances of subfamilies I, II, and III and other subfamilies were distant, while those of subfamilies XIV, XV, and XVI were relatively close. The C2H2 ZFPs of *B. rapa*, *B. napus*, and *Arabidopsis* were unevenly distributed in each subfamily. The phylogenetic homology of C2H2 transcription factors in *B. rapa*, *B. napus*, and *Arabidopsis* indicates that they may play the same or similar roles in some biological functions.

### 2.3. Gene Structure of the C2H2 ZFP Gene Family

Using MEME for conservative motif analysis, it was found that most of the genes distributed in the same evolutionary branch had the same motif (Figure 3B, Appendix A). Gene structure analysis, as shown in Figure 3C, and Appendix A, showed that most of the genes in this family have only one CDS. A number of stress-related response elements were identified, such as low temperature response elements, anaerobic response elements, and drought response elements. In addition, there were hormone-related response elements, such as IAA, SA, ABA, GA, and MeJA, indicating that the gene family has a variety of *cis*-acting elements and diverse functions (Figure 3D). The C2H2 family genes in *B. rapa* have a variety of functional elements related to hormonal and abiotic stress, suggesting that C2H2 family genes may regulate plant growth and development and a variety of stress responses through different hormones.

### 2.4. Genomic Collinearity Analysis of the C2H2 ZFP Gene Family

Intraspecific collinearity analysis of the C2H2 ZFP genes in *B*. *rapa* showed that 126 pairs of genes with collinear relationships, such as Bra000245 and Bra016900, Bra000245, Bra004660, Bra004660 and Bra003317, and Bra004660 and Bra007378, were distributed on 10 chromosomes (Figure 4A). This suggests that the C2H2 ZFP gene is highly conserved and under stabilizing selection, and may be produced by gene replication. It was found that there was a collinear relationship among the C2H2 ZFP genes in *B. rapa*, *Arabidopsis*, and *B. napus*, indicating that the gene family was highly conserved in the process of evolution (Figure 4B). The Ka/Ks of *B. rapa* and *B*. *napus* ranged from 0.044 to 0.755, with an average of 0.331. The Ka/Ks of *B. rapa* and *Arabidopsis* ranged from 0.072 to 0.54, with an average of 0.262. It is suggested that the C2H2 ZFP gene is relatively conserved in cruciferous species (Figure 4C, Appendix A).

### 2.5. Analysis of the Gene Expression Characteristics of Members of the Gene Family

The expression of 53 and 52 genes of the cold-resistant ‘Longyou-7’ was upregulated at 3 h and 24 h, respectively (Figure 5, Appendix A). Among them, 36 genes were upregulated at 3 h and 24 h under low-temperature stress, and 22 genes, including Bra000245, Bra001752, and Bra002528, were higher at 24 h than at 3 h. Additionally, the expression levels of 11 genes such as Bra000245, Bra002528, and Bra004312 in the highly cold-resistant ‘Longyou-7’ were higher than those in the weaky cold-resistant ‘Lenox’. In addition, there were 25 downregulated genes in ‘Longyou-7’ at 3 h and 24 h, among which 10 genes, such as Bra034396, Bra035609, and Bra024706, had lower expression levels than in ‘Lenox’. After 6 h of freezing stress, there were 16 upregulated genes, 62 downregulated genes, and 63 non-expressed genes in ‘Longyou-7’ and ‘Tianyou-2’. There were 21 upregulated genes, 51 downregulated genes, and 69 non-expressed genes in ‘Tianyou-2’. There were 22 genes expressed differently between ‘Longyou-7’ and ‘Tianyou-2’. Among them, seven genes had higher expression levels in ‘Longyou-7’ than in ‘Tianyou-2’, and 15 genes had higher expression levels in ‘Tianyou-2’ than in ‘Longyou-7’.

### 2.6. Analysis of the Expression Pattern of Gene Family Members

Ten genes differentially expressed under low-temperature stress were screened from the transcriptome results of two varieties differing in cold resistance. The expression patterns of these genes under cold, drought, high temperature, and salt stresses were further analyzed. The untreated rapeseed was used as a control (CK) for four abiotic stress treatments. The expression induced by Bra002528 and Bra004312 was more obvious under cold stress, in which the expression of Bra002528 in the growth cone of Bra002528 ‘Longyou-7’ reached 17 times that of CK and 5.5 times that of the ‘Lenox’ growth cone at 24 h. The expression of Bra002528 in the roots of ‘Longyou-7’ at 24 h was 24.7 times that of CK and 2.8 times that of ‘Lenox’, and the expression of the Bra004312 gene in ‘Longyou-7’ was 3 times higher than that in ‘Lenox’ at 24 h. The expression level of Bra004312 was 4.6 times higher than that in ‘Lenox’ at 24 h in the growth cone (Figure 6A). At 24 h of high-temperature stress treatment, the expression of Bra002528, Bra004312, and Bra020284 genes in ‘Longyou-7’ leaves was 2.8, 2.5, and 3 times higher than that in ‘Lenox’ leaves. Expression in stems was 1.2, 2.5, and 3.1 times higher than that in ‘Lenox’. Expression in roots was 4.2, 4.2, and 3 times higher than in ‘Lenox’ (Figure 6B).

Under drought stress, the expression of Bra002528 and Bra006691 was more significantly induced by drought. The expression of Bra002528 and Bra006691 in the leaves of ‘Longyou-7’ at 24 h was 7.9 and 14.4 times that of CK and 3.3 and 8.2 times that of ‘Lenox’, respectively, while in the growth cone, the expression of Bra002528 and Bra006691 in ‘Longyou-7’ at 24 h was 20.4 and 24.8 times that of CK and 5.5 and 13.4 times that of ‘Lenox’, respectively. The expression levels of these genes in the root of ‘Longyou-7’ at 24 h was 25.8 and 10.5 times that of CK, respectively, and 5.6 and 5.7 times that of ‘Lenox’, respectively (Figure 6C). After 24 h of salt stress, the expression of Bra004660 in the ‘Longyou-7’ growth cone was 26.5 times higher than that in CK, and 6.6 times higher than that in ‘Lenox’. The expression of Bra020284 in the ‘Longyou-7’ growth cone was 27.1 times higher than that in CK, and 6.9 times higher than that in ‘Lenox’ (Figure 6D).

With these results we were surprised to find that Bra002528, Bra004312, Bra004312, Bra020284, Bra006691, and Bra004660 genes were expressed at higher levels in ‘Longyou-7’ than ‘Lenox’ after stress, and that the expression was higher in growth cones and roots, followed by leaves and stems. In addition, we identified Bra002528, Bra020284, and Bra004312 genes in response to cold- and high-temperature stress, Bra006691 gene in response to drought stress, and Bra004660 gene in response to salt stress. In addition, individual genes were found to respond to multiple abiotic stresses, e.g., Bra002528 gene responded to cold, high-temperature and drought stress; Bra004312 responded to cold and high-temperature stress; Bra020284 responded to high-temperature and salt stress.

### 2.7. Screening and Identification of Transgenic Arabidopsis with the BrZAT12 Gene

After infesting *Arabidopsis*, the plants were screened using Basta for three generations, and the T3 generation of *Arabidopsis* plants exhibiting normal growth was transplanted into pots for the next experiment (Appendix A). The PCR amplification results showed that wild-type *Arabidopsis* plants had no amplified bands, and 12 of the randomly selected transgenic plants from Z1 to Z13 showed a 483 bp band after agarose gel electrophoresis, while the Z3 plants failed to show a band. The expression of the *BrZAT12* gene in transgenic *Arabidopsis* was higher than that in wild-type *Arabidopsis* and was significantly different from that in wild-type *Arabidopsis* (Appendix A). The gene expression of Z5 plants was the highest, which was 4.65 times higher than that of wild-type *Arabidopsis*. The results showed that all the six plants that over-expressed *BrZAT12*, and Z5 had the highest transformation efficiency (Appendix A).

### 2.8. Phenotypic and Expression Analysis of BrZAT12-Transformed Plants under Low-Temperature Stress

It was found that the leaves of the wild-type and transgenic *Arabidopsis* plants with the *BrZAT12* gene exhibited no significant change after 3 h of treatment, and the survival rate was similar. After 6 h of treatment, the leaves of the plants began to wilt, the survival rate of wild-type *Arabidopsis* decreased by about 83.3%, and the survival rate of transgenic plants with the *BrZAT12* gene was 91.7% (Figure 7A,B). After 12 h of treatment, the wild-type plants froze, wilted, and died after being returned to normal temperature for a week, and the survival rate was just 16.7%, while the survival rate of the transgenic *Arabidopsis* was 62.5%, which was significantly different from that of the wild-type. The survival rate of wild-type *Arabidopsis* treated for 24 h was 4.17%, and most of the transgenic plants with the *BrZAT12* gene returned to normal growth at room temperature, and the survival rate reached 41.7%, which was significantly different from that of the wild-type (Figure 7C). The expression of the *BrZAT12* gene in transgenic *Arabidopsis* plants reached the highest level after 12 h of treatment, which was 18.8 times higher than that in the wild-type *Arabidopsis* plants treated for 12 h and was significantly different from that in the wild-type *Arabidopsis* plants treated for 12 h (Figure 7D). It is suggested that the *BrZAT12* gene can improve the cold resistance of plants.

### 2.9. Physiological-Activity Analysis of the BrZAT12 Gene in Transgenic Plants at Low Temperature

After 24 h of low-temperature treatment, the SOD enzyme activity of the transgenic *Arabidopsis* was 3.17 times higher than that of the control and 1.90 times higher than that of the wild-type plants treated for 24 h. The activity of SOD in the wild-type plants reached the maximum at 24 h, which was 1.54 times higher than that of the untreated plants (Figure 7E). Regarding the POD enzyme activity, the enzyme activity of the transgenic *Arabidopsis* plants reached the highest at 24 h of treatment, which was 2.54 times higher than that of the untreated *Arabidopsis* and 1.42 times that of the wild-type *Arabidopsis*. There was a significant difference in POD activity between the transgenic *Arabidopsis* and wild-type *Arabidopsis* at 24 h (Figure 7C). The enzyme activity of the wild-type *Arabidopsis* was 1.91 times higher than that of the untreated *Arabidopsis* for 24 h. The CAT activity of the transgenic *Arabidopsis* was 2.13 times higher than that of the untreated *Arabidopsis* and 1.38 times that of the wild-type *Arabidopsis* treated for 24 h. The CAT activity of the wild-type *Arabidopsis* increased by 1.61 times as much as that of the untreated *Arabidopsis* at 24 h, and the difference was significant (Figure 7G). The activities of the three enzymes in the transgenic plants were higher than those in the wild-type plants. The results showed that the reactive oxygen species (ROS) scavenging ability of the transgenic plants with the *BrZAT12* gene was stronger than that of the wild-type plants.

### 2.10. The BrZAT12 Gene Is Involved in the Regulation of ABA and MAPK Signaling Pathways

The results showed that the ascorbate peroxidase (APX) activity and *BrZAT12* gene expression of ‘Longyou-7’ and ‘Lenox’ were significantly lower than those of CK after treatment with PD98059 at room temperature. The SOD and APX activities of ‘Longyou-7’ and ‘Lenox’ were 1.46 and 1.22 times higher than those of CK, respectively, while the SOD and APX activities of ‘Lenox’ were 1.18 and 1.10 times higher than those of CK, respectively. The expression of *BrZAT12* in ‘Longyou-7’ was significantly higher than that in ‘Lenox’. The SOD and APX activities of ‘Longyou-7’ under low-temperature treatment were 1.22 and 1.21 times higher than those of CK, respectively, and those of ‘Lenox’ were 1.08 and 1.10 times higher than those of CK, respectively. The gene expression and enzyme activity of *BrZAT12* after co-treatment were higher than those of the PD98059 treatment, but were lower than those of the control. Compared with the ambient treatment, the expression changes in ‘Longyou-7’ under low-temperature treatment were relatively small, indicating that ‘Longyou-7’ was less affected by the inhibitor PD98059 and exogenous ABA under low-temperature stress (Figure 8). It is speculated that the *BrZAT12* gene may play a role in the involvement of MAPKK in ABA signaling processes.

## 3. Discussion

ZFP genes are abundant in plants, and they play an important role in plant growth and development and in regulating the plant response to various biotic and abiotic stresses [25,26]. Ninety-nine C2H2 ZFP genes were identified in tomato and were classified into four groups, and 32 C2H2 genes were identified in response to multiple stresses. Pathway analysis identified five gluconeogenic and sphingolipid-related pathways and endocytosis pathways as being enriched [18]. In this study, we identified, expressed, cloned, and analyzed the C2H2 ZFP gene family in *B. rapa.* A total of 141 members of the C2H2 gene family were identified, which were distributed on 10 chromosomes. Most genes in the same evolutionary branch contained the same conserved motif, which was similar to that of *B. rapa*. Collinear relationship analysis showed that the gene was highly conserved in the process of evolution. In addition, 77 and 246 members of the ZFP family were identified in *Arabidopsis* and *B. napus*, respectively (Figure 2). In view of the fact that genome duplication occurred about 4 million years after the *Brassica* genome split from *Arabidopsis* ancestors, the doubling of these ZFPs in diploid *Brassica* plants reflects genome duplication [27]. About 7500 years ago, the genomes of *B. oleracea* and *B. rapa* also fused to form *B. napus*, which led to a doubling of the number of ZFPs [27]. It is generally believed that this increase in gene expression will cause an imbalance in gene expression and, over time, lead to a decrease in excessive gene expression.

An earlier study identified a total of 146 Q-type C2H2-ZFPs, including 37 in *B. oleracea*, 35 in *B. rapa*, and 74 in *B. napus*. The sequence similarity and alignment of these genes on the chromosome remained essentially unchanged in *B. napus* compared to those inherited in *B*. *oleracea* and *B. rapa*. By contrast, protein sequences differed more between homologous genes in *B. oleracea* and *B. rapa* and were organized more differently on the chromosomes. Overall, these 146 proteins are highly conserved, especially in known motifs [28]. A search of the cabbage genome identified 25 proteins with 2-Q zinc-finger structures. Furthermore, a comparative map position analysis of *Arabidopsis* and *B. oleracea* genes revealed that *Arabidopsis* inherited all 24 blocks, but the final karyotype was reorganized and reduced to five chromosomes. These homozygous blocks are found in modern *Brassica* species such as *B. rapa* and *B. oleracea*. These results further support the idea that the *B*. *oleracea* genome originated from ancient polyploids [29,30,31].

Under different stress treatments, the Bra002528, Bra004312, Bra020284, Bra006691, and Bra004660 genes were significantly expressed in the root and growth cone of ‘Longyou-7’, indicating that they play a certain role in the stress response of *B. rapa*. Among them, the Bra002528 gene was significantly expressed in the growth cone and root of the cold-resistant *B. rapa* ‘Longyou-7’ and was significantly expressed in different stress environments, indicating that the Bra002528 (*BrZAT12*) gene plays an important role in regulating the response of *B. rapa* to stress [32,33]. After the *TaZNF* gene was transferred into *Arabidopsis*, it was found that the chlorophyll content, proline content, and stomatal reduction of the transgenic *Arabidopsis* were higher than those of the wild-type *Arabidopsis* under salt stress, and the salt resistance of the transgenic plants was enhanced [34]. C2H2 ZFPs enhance cold resistance by directly regulating cold-related downstream genes in plants. Transgenic *Arabidopsis* and tobacco (*Nicotiana tabacum*) overexpressing *SCOF-1* showed increased expression of cold-regulated (COR) genes and enhanced cold tolerance. The *SCOF-1* transgenic plants recovered from cold stress faster than the control, and the T2 generation *SCOF-1* transgenic *Arabidopsis* still expressed more COR genes, such as *COR15a*, *COR47*, and *RD29B*, than the wild type at normal growth temperatures. SCOF-1 significantly enhanced the binding activity of *SGBF-1* to the ABRE sequence in vitro, thereby promoting COR gene expression and enhancing cold tolerance. The interaction of these two proteins is required for the role of ABRE in COR gene expression to enhance cold tolerance [3,35,36]. In addition, SCOF-1 enhanced cold tolerance in transgenic sweet potato and transgenic potato [37,38].

It was shown that the C2H2 ZFPs could enhance plant cold resistance by increasing the level of osmotic substances. *ZFP182* significantly enhanced cold tolerance in rice overexpression lines by increasing the expression of *OsP5CS* and *OsLEA3* and the accumulation of osmoprotectants [39]. *GmZF1* enhanced cold tolerance in transgenic *Arabidopsis* by increasing proline and soluble sugar contents and reducing membrane lipid peroxidation under low-temperature stress [22]. In this study, after low-temperature treatment of the transgenic *BrZAT12* plants, it was found that there was less wilting of, and damage to, the transgenic *Arabidopsis* leaves compared to the wild-type *Arabidopsis*, and the survival rate of the transgenic *Arabidopsis* was significantly higher than that of the wild-type *Arabidopsis*, indicating that the *BrZAT12* gene did improve the cold resistance of the plants. The enzyme activities of both the wild-type and *BrZAT12* transgenic plants increased after low-temperature treatment, and the increasing trend of enzyme activities in the *BrZAT12* transgenic *Arabidopsis* plants was higher than that in the wild-type plants. The results indicated that the *BrZAT12* gene was associated with cold resistance, and the transgenic *Arabidopsis* with the *BrZAT12* gene demonstrated improved cold resistance [40].

C2H2 ZFPs play an important role in plant tolerance to low temperatures through ABA-dependent and ABA-independent pathways [13,24]. In addition to the ABA signaling pathway, C2H2 ZFPs enhance abiotic stress resistance through the MAPK signaling pathway [7,11]. Thus, C2H2 ZFPs regulate abiotic stress responses through the ABA signaling pathway and MAPK signaling pathway, constituting an interactive and complex regulatory network [20,41]. It was reported that the overexpression of *ZFP245* improved tolerance to drought and cold damage in rice. In addition, the overexpression of *ZFP245* promoted rice resistance to H_2_O_2_ and increased the content of osmoregulatory substances. This indicates that *ZFP245* can enhance tolerance to abiotic and oxidative stresses through proline biosynthesis and scavenging of ROS. Importantly, the overexpression of *ZFP245* enhanced plant sensitivity to exogenous ABA, suggesting that ZFP245 is likely involved in abiotic stress responses through an ABA-dependent signaling pathway [42,43]. The overexpression of *ZFP36* increased the expression level and activity of SOD and APX, which in turn enhanced tolerance to water and oxidative stresses. Meanwhile, ZFP36 regulates the expression level of NADPH oxidase and H_2_O_2_ production in the ABA signaling pathway [44,45].

ZAT12 is involved in abiotic-stress regulation through ABA non-dependent signaling pathways and the MAPK signaling pathway as an active repressor [46,47,48]. EIN3, which is phosphorylated by MPK3/6, can directly upregulate *ZAT12* expression, indicating that ZAT12 may be involved in the MAPK signaling pathway [49,50,51]. In addition, the expression of *ZAT12* and *ZAT7* preceded the increased expression of *APX1* in response to oxidative stress. The expression of *ZAT12* and *ZAT7* in the loss-of-function mutant *apx1* increased with increasing levels of H_2_O_2_ [52,53]. Another study showed that *CBF* is also regulated by ZAT12 and acts upstream of ZAT10 and AZF1/2/3 through the MAPK signaling pathway. Thus, different ZAT proteins may be highly related and involved in responses to abiotic stresses in a similarly coordinated and cooperative manner [20,54,55]. The results showed that the activities of SOD and APX and the expression of the *BrZAT12* gene in plants increased after exogenous ABA treatment, and the increased trend in enzyme activity of ‘Longyou-7’ was greater than that of ‘Lenox.’ However, the inhibitor treatment of MAPKK could decrease the enzyme activity and expression. The sensitivity of plants to ABA and inhibitors decreased at low temperature. After the combination of exogenous ABA and inhibitors, the enzyme activity and expression were still lower than those in the control group. It is speculated that the *BrZAT12* gene may play a role in the ABA signaling process in which MAPKK is involved.

## 4. Materials and Methods

### 4.1. Plant Growth and Treatments

Two *B. rapa* varieties ‘Longyou-7’ (ultra-cold resistant, AA, 2n = 20), ‘Tianyou-2’ (cold resistant, AA, 2n = 20), and ‘Lenox’ (weakly-cold resistant, AA, 2n = 20) with differential cold resistance were selected. ‘Longyou-7’ and ‘Lenox’ seeds of uniform size and healthy fullness were selected, sterilized using 5% NaClO solution, rinsed with distilled water after sterilization, and placed evenly on filter paper for germination. After the seeds germinated, they were sown in pots. Four seeds were sown in each pot, and the pots were incubated in a light incubator (14 h light, 25 °C; 10 h dark, 20 °C) until the seven-leaf stage. The untreated rapeseed was used as a control (CK) for four abiotic stress treatments: low-temperature treatment (in a 4 °C incubator), high-temperature treatment (in a 40 °C incubator), drought treatment (simulating drought stress using 18% polyethylene glycol solution), and high salt treatment (simulating salt stress using 180 mmol/L NaCl solution). Four parts, namely, the leaves, roots, stems, and growth cones, were used for subsequent experiments at 0, 1, 3, and 24 h after various treatments. Another set of experiments was conducted to study the involvement of the *BrZAT12* gene in the regulation of MAPK and ABA signaling. The seeds of ‘Longyou-7’ and ‘Lenox’ were cultured in MS medium for 1 week and then transferred to four groups of different treatment media for 10 days as follows: control group, MS medium; ABA treatment group, MS medium + ABA (100 μM); MAPKK inhibitor treatment group, MS medium + PD98059 (MAPKK kinase specificity inhibitor) (50 μM); ABA + MAPKK inhibitor treatment group, and MS medium + PD98059 (50 μM) + ABA (100 μM). The above samples were snap frozen in liquid nitrogen and subsequently stored in a refrigerator at −80 °C, followed by RNA extraction and reverse transcription, with three biological replicates set up for each treatment.

### 4.2. Identification of Members of the C2H2 ZFP Gene Family

In this study, *B. rapa* genome and gene structure annotation information with high homology were downloaded from the Brassica Database (http://brassicadb.org/brad/index.php/ (accessed on 13 March 2022)). The C2H2 ZFP domain (PF13912) HMM was downloaded from the Pfam (http://pfam.xfam.org/family/PF13912 (accessed on 13 March 2022)) [56,57]. Blast was used to screen the members of the C2H2 ZFP family in *B. rapa*. The CDD online tool (http://www.ncbi.nlm.nih.gov/Structure/bwrpsb/bwrpsb.cgi (accessed on 13 March 2022)) was used for further screening [28,58]. TBtools software (https://github.com/CJ-Chen/TBtools (accessed on 14 March 2022)) was used to determine the position of the C2H2 ZFP gene family members on chromosomes in *B. rapa*, and visual analysis was carried out [59].

### 4.3. Analysis of Members of the C2H2 ZFP Gene Family

Conservative motif analysis was carried out on the MEME (http://meme-suite.org/tools/meme (accessed on 17 March 2022)), and the number of motifs identified was set to 10 [60]. The upstream 2000 bp sequence of the gene initiation codon was selected by TBtools and submitted to PlantCare (http://bioinformatics.psb.ugent.be/webtools/plantcare/html/ (accessed on 17 March 2022)) for *cis*-acting element analysis [61]. The evolutionary tree, motifs, gene structures, and *cis*-acting elements of the C2H2 gene family members of *B. rapa* were combined and visually analyzed using TBtools software. The C2H2 ZFP sequences of *Brassica napus* and *Arabidopsis* were downloaded from the PlantTFDB database (http://planttfdb.cbi.pku.edu.cn/ (accessed on 13 March 2022)) [62,63]. A phylogenetic tree was constructed using MEGA7.0 software (neighbor-joining, NJ) with the bootstrap value set to 1000, and the tree was edited using the Evolview online website (http://www.evolgenius.info/evolview/ (accessed on 19 March 2022)) [64,65].

### 4.4. Collinear Analysis of Gene Family Members

The collinearity of members of the C2H2 gene family in *B. rapa* was analyzed using the Amazing Super Circos tool of TBtools. The collinear relationship between *B. rapa*, *B. napus*, and *Arabidopsis* was visually analyzed by TBtools software. The Ka and Ks replacement rates of *B. rapa*, *B. napus*, and *Arabidopsis* were calculated using TBtools’ Calculator tool [59]. MapChart software was used to map the chromosomal location of the C2H2 gene family [66].

### 4.5. RNA Isolation, Reverse Transcription, qRT-PCR, and Transcriptome Expression Analysis

RNA Extraction Kits (TIANGEN Biotech Co., Ltd., DP432, Beijing, China) were used to extract total RNA from ‘Longyou-7’ and ‘Lenox’ and remove genomic DNA contamination. The cDNA was obtained using the PrimeScript™ RT Master Mix (Takara, RR036A, Beijing, China) according to the instructions. The cDNA was checked for quality and concentration using an ultra-trace UV-visible spectrophotometer (NanoVueTM Plus, Wilmington, DE, USA) and kept on standby. Ten differentially expressed genes in the transcriptome data were selected, and specific primers were designed for qRT-PCR (ABI QuantStudio 5, Thermo Fisher Scientific, Shanghai, China) under low-temperature, high-temperature, salt, and drought stress (Appendix A) according to the instructions of the TB Green^®^ Premix Ex Taq™ II (Takara, RR820A, Beijing, China). The reaction process was as follows: 5 min at 95 °C, followed by 40 cycles of 5 s at 95 °C and 30 s at 60 °C, followed by 65–95 °C melting curve detection. The qRT-PCR efficiency of the genes was obtained by analyzing the standard curve of the cDNA gradient dilution. The gene fragment encoding *B. rapa β-actin* RNA was used as an internal control to normalize the amount of template cDNA. The relative expression values of each gene were calculated using the comparative 2^−ΔΔCT^ method [67]. The RNA-seq libraries (SRP179662) and (SRP211768) were selected for gene-expression analysis [68,69]. Three biological replicates were set up for each experimental treatment.

### 4.6. Preliminary Verification of BrZAT12 Function

The *BrZAT12* (Bra002528) gene of *B. rapa* was cloned by PCR based on specific primers (*BrZFP1*-F: ATGGTTGCTATTTCAGAGATCAAGTCGACG; *BrZFP1*-R: ACAAACAGGCCTTCCAAGTTC). With reference to Gateway technology, PCR amplification was carried out using primers (*BrZAT12*-vec-F: AAAAAAGCAGGCTTCATGGTTGCTATTTCAGAGATCAAGTCGACG; *BrZAT12*-vec-R: AGAAAGCTGGGTCACAAACAGGCCTTCCAAGTTC), and the recovered products were ligated with BP Clonase enzyme (Invitrogen, Carlsbad, CA, USA) and the pDONR vector. After transforming into coliform bacteria (DH5α), the positive clones were screened by smear sequencing. The positive clones were selected to interact with LR Clonase enzyme (Invitrogen, Carlsbad, CA, USA) to connect to the overexpression vector pEarly-Gate101, following the instructions for *Agrobacterium tumefaciens* GV3101 (Anyu Biotechnology Co., Ltd., Shanghai, China). After the bacteria were picked out from the coated plate, colony PCR was carried out and the positive bacterial solution was selected and preserved.

Transgenic *Arabidopsis* was obtained by the floral-dip method. The T3 seeds were obtained by screening by spraying with 0.01% Basta (10%) solution. The T3 seeds of transgenic *Arabidopsis* lines and wild-type *Arabidopsis* were sown in pots according to the experimental requirements. After low-temperature treatment in the incubator at −4 °C for 0, 3, 6, 12, and 24 h, one part was subjected to RNA extraction and physiological index determination, and the other part resumed normal growth, and the plant phenotype and survival rate were observed and counted after 1 week. Primers (BrZAT12-pcr-F: GCGAACTGTCTGATGCTC; BrZAT12-pcr-R: TTAGGCTTCTTGTGGCTC) were used to perform qRT-PCR on *Arabidopsis.* qUBC (F: ACAGCGAGAGAAAGTAGCAGA; qUBC-R: TTGATAAGAGCGGTCCATTTGAA) was used as an internal control for *Arabidopsis.* The CAT activity was calculated by measuring the decrease in absorbance at 240 nm with H_2_O_2_. POD activity and SOD was estimated according to the method developed by Gill [70,71]. Three biological replicates were set up for each of the above experimental treatments.

## 5. Conclusions

In this study, the C2H2 ZFP gene family was found to be distributed on 10 chromosomes of *B*. *rapa*. The genes were highly conserved during evolution. Quantitative analysis revealed that the Bra002528 gene was significantly expressed in the growth cones and roots of the cold-resistant rapeseed ‘Longyou-7’ as well as in different stress environments, indicating that this gene plays an important role in plant growth and development and the stress response. The survival rate of the trans-*BrZAT12* gene in *Arabidopsis* was higher than that in the wild-type at low temperature, and the degree of leaf damage was lower than that in the wild-type. The activities of SOD, POD, and CAT were higher in the transgenic *Arabidopsis* than in the wild-type *Arabidopsis.* The expression of the *BrZAT12* gene was induced by low temperature and played a role in the ABA signaling process involved in MAPKK, and it could enhance the cold resistance of plants. These results provide a theoretical basis for analyzing the cold-resistance mechanism of winter *B. rapa* and for increasing the cold resistance of northern winter rapeseed.

## Figures and Tables

**Figure 1 ijms-23-12218-f001:**
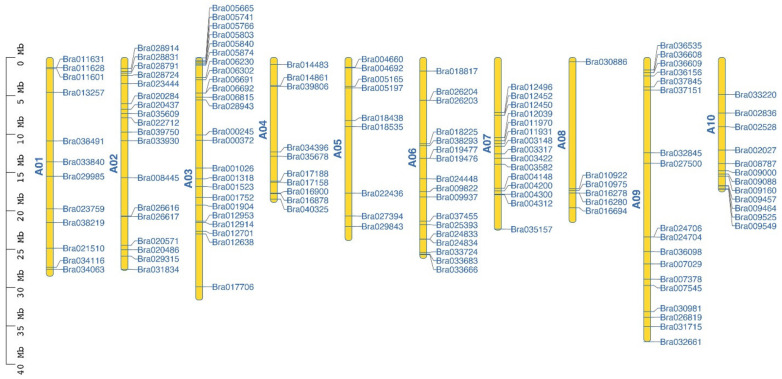
Chromosome distribution of C2H2 ZFP family members in *B. rapa*. The 141 members are distributed over 10 chromosomes.

**Figure 2 ijms-23-12218-f002:**
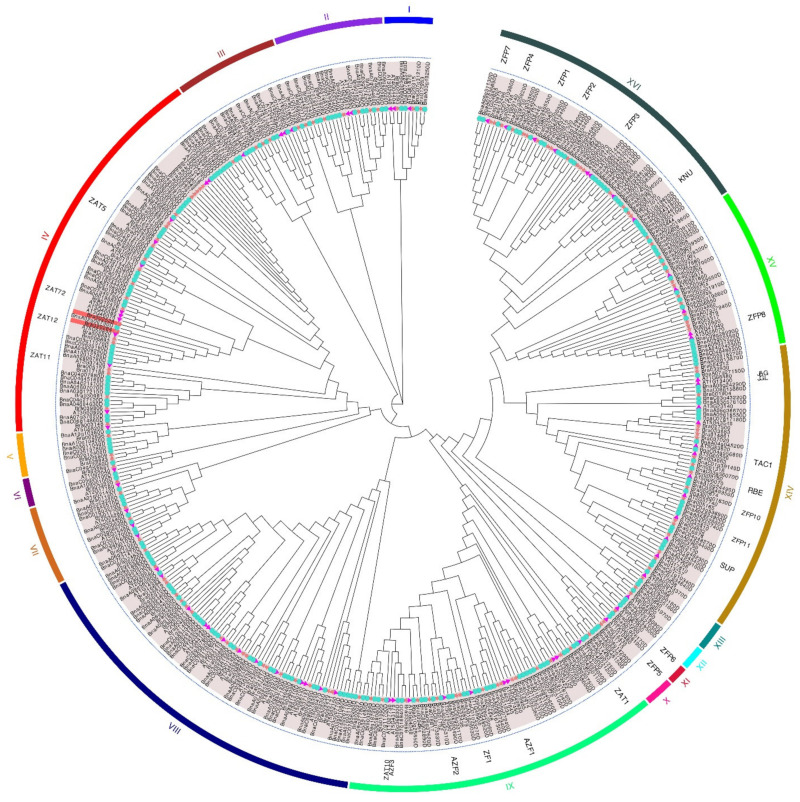
Phylogenetic tree of the C2H2 ZFP family. The 16 subfamilies are colored differently. Red stars represent *B. rapa*, blue circles represent *B.*
*napus,* and purple triangles represent *Arabidopsis.* The abbreviations indicate the ZFP gene family known in *Arabidopsis.*

**Figure 3 ijms-23-12218-f003:**
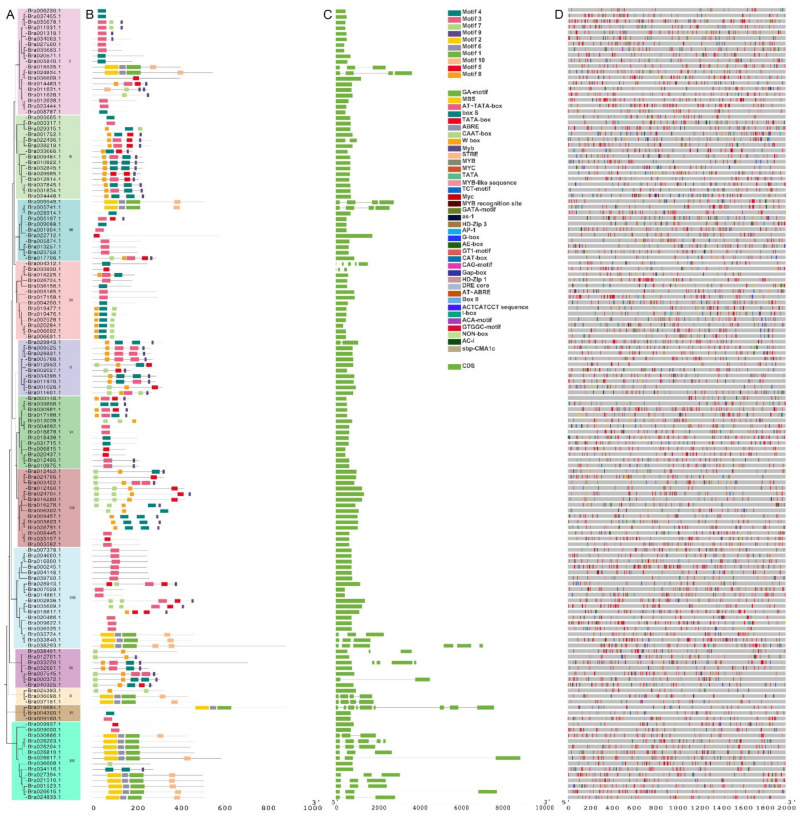
Analysis of the gene structure of the C2H2 ZFP gene family in *B. rapa*. (**A**) The 141 members are assigned to 12 sub-clusters. (**B**) Each colored box represents a motif, and the gray line represents a non-conserved sequence. (**C**) Exons and introns of each subgroup are indicated by green boxes and grey lines, respectively. (**D**) The specific colored boxes indicate the different *cis*-acting elements.

**Figure 4 ijms-23-12218-f004:**
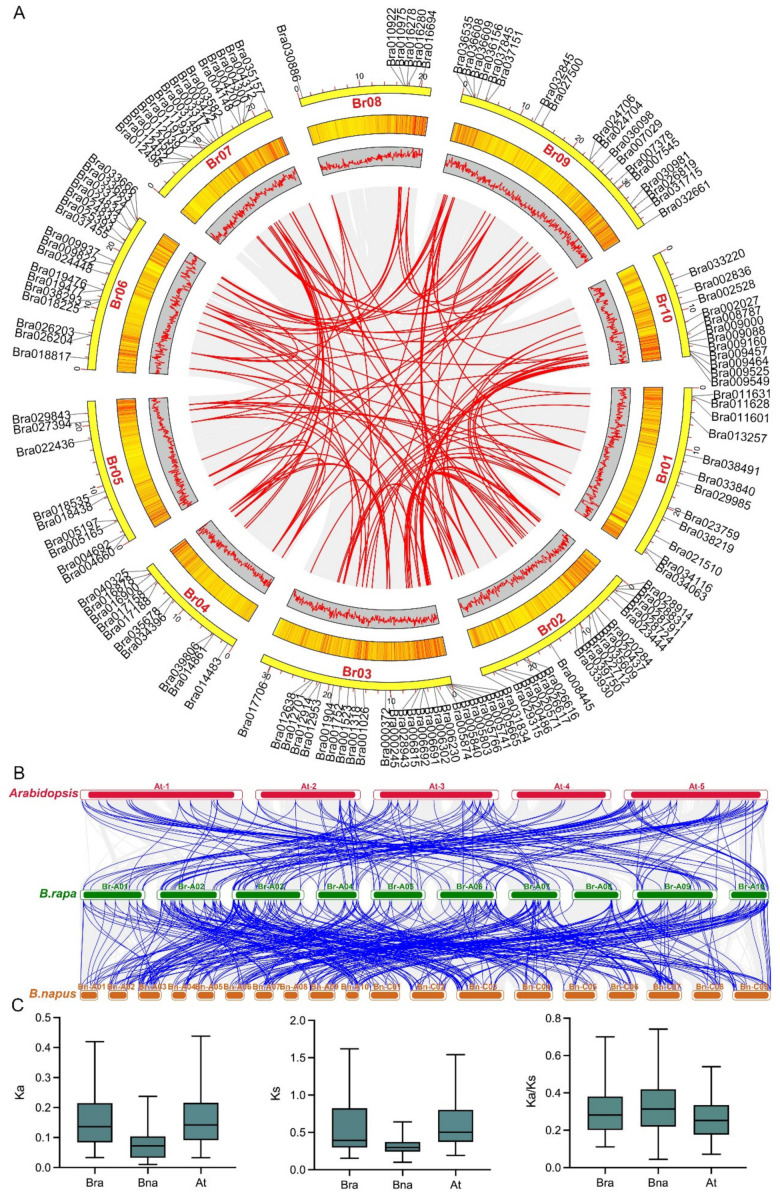
C2H2 ZFP gene family collinearity analysis. (**A**) The gene names are marked outwards on the corresponding chromosome. Duplicate gene pairs are highlighted with red lines. (**B**) The red, green, and orange indicate the chromosomes of *Arabidopsis*, *B. rapa*, and *B. napus,* respectively. Duplicate gene pairs are highlighted with blue connecting lines. (**C**) Ka, Ks, and Ka/Ks values of homologous gene pairs among species.

**Figure 5 ijms-23-12218-f005:**
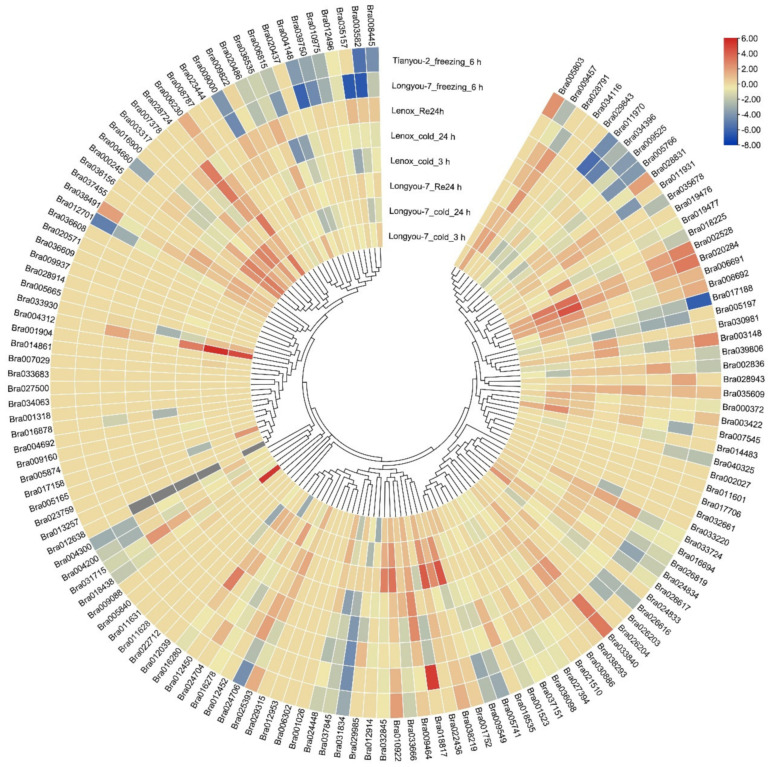
Expression of C2H2 ZFP genes in different varieties under cold and freezing stress. Heat maps were expressed using log2 values for each gene. The color scale represents the relative expression levels from low (blue) to high (red).

**Figure 6 ijms-23-12218-f006:**
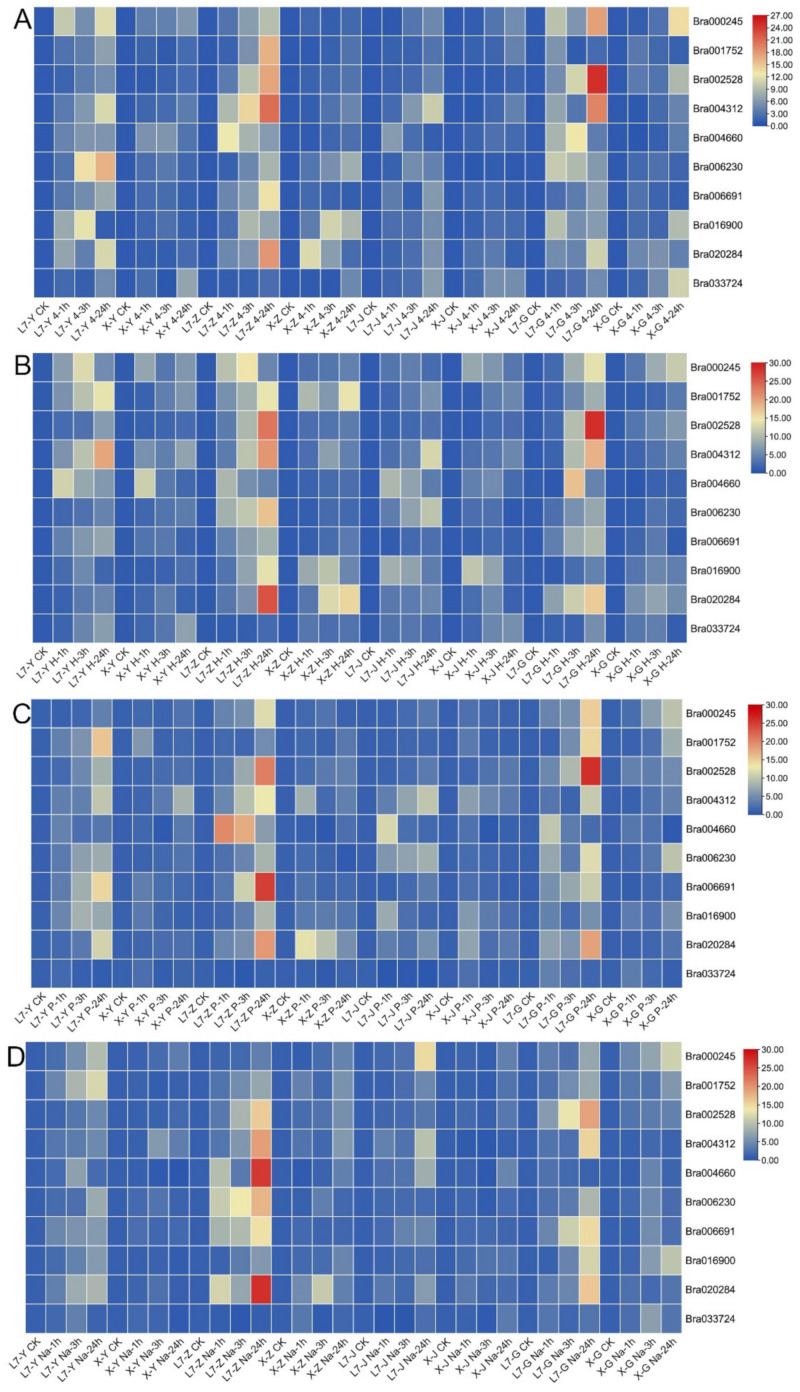
Heatmap of the expression of C2H2 ZFP genes in *B. rapa* under abiotic stress. (**A**–**D**) represent cold, high temperature, drought, and salt stress, respectively. Y, Z, J, and G in the column labels represent leaf, growth cone, stem, and root tissues of rapeseed, respectively.

**Figure 7 ijms-23-12218-f007:**
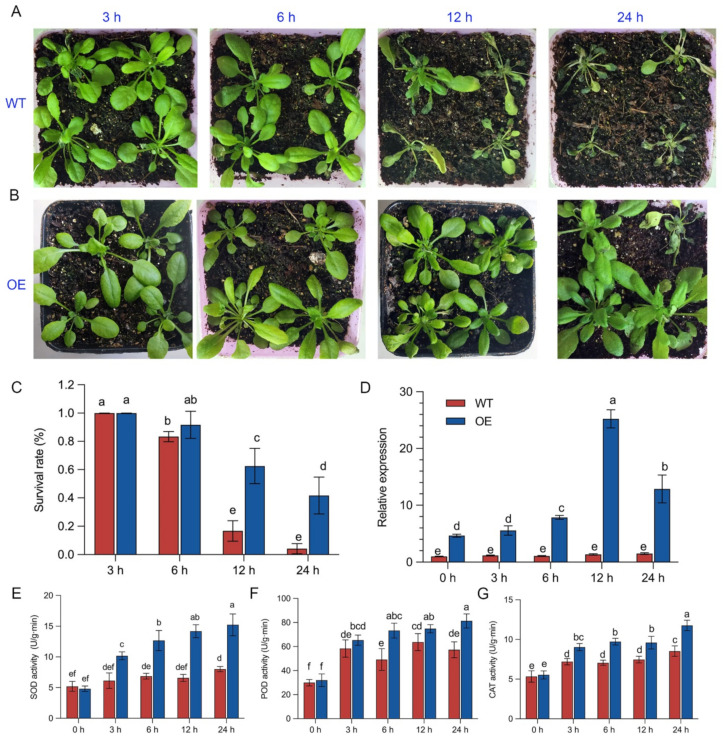
Phenotype, expression, and physiological activity of BrZAT12 transgenic *Arabidopsis* after low-temperature treatment. (**A**,**B**) Phenotype of *BrZAT12* transgenic *Arabidopsis* after low-temperature treatment. (**C**) Survival rate of *Arabidopsis* plants after low-temperature treatment. (**D**) Quantitative RT-PCR test of *BrZAT12* transgenic *Arabidopsis* at low temperature. (**E**–**G**) SOD, POD, and CAT activities, respectively. WT: wild-type *Arabidopsis*, OE: *BrZAT12* transgenic *Arabidopsis*. a–f represent significant differences between treatments at *p* ≤ 0.05.

**Figure 8 ijms-23-12218-f008:**
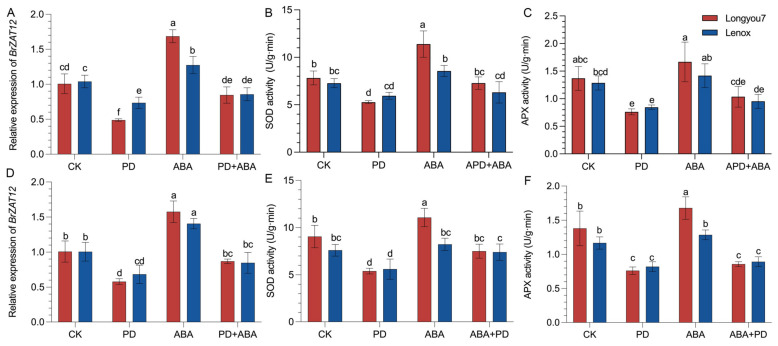
Expression of *BrZAT12* and activities of SOD and APX in *B. rapa* under different treatments. (**A**–**C**) Room-temperature treatment and (**D**–**F**) low-temperature treatment. a–e represent significant differences between treatments at *p* ≤ 0.05.

## Data Availability

Data is contained within the article and Appendix A.

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
