# Peer review of "Genome-Wide Identification of C2H2 ZFPs and Functional Analysis of BRZAT12 under Low-Temperature Stress in Winter Rapeseed (Brassica rapa)"

_ijms, 2022, doi:10.3390/ijms232012218_

Round 1
Reviewer 1 Report
The manuscript consolidates the set of identified C2H2 ZFP transcripton factors from B. rapa, B. napus and A. thaliana. Sequences are compared using phylogenetic analysis. Sequences are further examined for B. rapa sub-clusters and it was found that the members of clusters tend to contain the same motifs. This is not surprising because these clusters will tend to group proteins with similar functional groups. A co-linearity analysis was done on the ZFPs between B. rapa, B. napus and A. thaliana. It was concluded that these genes remain highly conserved. The authors next turn to a transcriptomic analysis using B. rapa where several different cultivars (including high and low temperature sensitivity lines) were treated with various abiotic stresses and transcriptomes were profiled after a treatment time course. The Authors identify several ZFPs that are induced highly in the resistant variety but not the cold susceptible variety. The strongest one (BrZAT12) was cloned and transformed into arabidopsis. These transgenic plants were shown to be significantly more resistant to freezing and a have higher induction of reactive oxygen scavenging enzymes upon cold treatment. Lastly, the authors examine levels of BrZAT12 as well as SOD and APX enzyme activity after inhibiting MAPKK pathway as well as ABA treatment. They see significant changes in response to both treatments and conclude that BrZAT12 is somehow involved with the MAPKK signaling and ABA response pathways.
My opinion is that this manuscript should be accepted for publication after revisions entirely related to clarification of the text and minor revisions of some figures.
I believe the transcriptomic-based identification of the BrZAT12 (as well as several other ZFPs) is of significance. The transcriptomic resources (RNA-seq data) generated in this study is substantial and would also be of general use to the Brassica community. In addition, the authors have a very convincing biological validation by transforming BrZAT12 into arabidopsis. These two aspects of the paper warrant publication by themselves.
I believe all other content presented here is appropriate for publication but has much less impact. The results associating BrZAT12 with MAPKK and ABA are interesting but would require further investigation to make any strong conclusion. However, I am not opposed to presenting these results as part of this publication.
Major Issues
Overall, I think the figures were well organized but had some minor labeling issues. The figure legends have major issues with not including sufficient descriptions of the figures. Also, the text needs to be re-organized. Throughout the paper, many sentences are difficult to follow and the exact meaning is not clear one example is listed below but this occurs throughout the manuscript and requires significant revisions. For much of the results section, the results are not prefaced by any explanation. This fact, combined with poor figure legends and some inadequate figure labeling make it very difficult to determine what the authors are intending to show with the data. The authors should not try to describe all the details of the figures in the text. The figures speak for themselves. The text should be used to point out the relevant information that you want to discuss about the figures and to show the parts of the results/figures that back up your claims. Much of the results section is simply stating statistics that are shown in the figures.
Minor Issues
Authors must explain what PD98059 is
Fig 4c – Ka/Ks analysis all ended up below 1. If you want to say anything, you need to determine the significance in your specific case but at face value, <1 means that these are under stabilizing selection and not under pressure to change/evolve.
The Authors state (line 150): “It is suggested that the C2H2 ZFP gene is highly conserved and consistent in the process of evolving” I’m interpreting this as the authors saying that the genes are under positive selections (in the process of evolving). I don’t think this is true based on Ka/Ks ratio. Please clarify this statement.
Section 2.5 text:
The authors should include a description of the cultivars and the logic behind the treatments done and tissues sampled. For instance, the Tianyou cultivar does not appear anywhere else in the manuscript and it is not clear what the purpose of this variety is.
Figure 5 is an interesting way to show the data. The gene expression appears to be clustered by phylogenetic relationships between the TFs. However, unless this arrangement is used to show something useful, it does not make sense to present data in this way. Please include text explaining the significance of arranging the data by phylogeny. If not, it may be more useful to view the data clustered based on expression level across cultivars (similar to Figure 6)
The abbreviation of “CK” is used for the untreated control. The authors need to define this abbreviation in the main text before it is used.
Figure 6: Please label the different tissues used in the heatmaps. The authors use “Y”,”Z”,”J” and “G” in the column labels. I assume these refer to different tissues. It is not clear which tissues.
Figure 8: Please label the bar colors. I’m assuming they refer to the different cultivars, but this is not shown anywhere.
A and D “Relative Expression” of which gene?
Figure 8A,D, text claims “The expression of BrZAT12 in ‘Longyou-7’ was significantly higher than that in ‘Lenox.’” The figure indicates that expression levels are the same. This also disagrees with previous data at least after the cold treatment.
Please mention (in the Materials and Methods) the number of replications done for each experiment.
Author Response
Response to Reviewer 1 Comments
Point 1: Major Issues
Overall, I think the figures were well organized but had some minor labeling issues. The figure legends have major issues with not including sufficient descriptions of the figures. Also, the text needs to be re-organized. Throughout the paper, many sentences are difficult to follow and the exact meaning is not clear one example is listed below but this occurs throughout the manuscript and requires significant revisions. For much of the results section, the results are not prefaced by any explanation. This fact, combined with poor figure legends and some inadequate figure labeling make it very difficult to determine what the authors are intending to show with the data. The authors should not try to describe all the details of the figures in the text. The figures speak for themselves. The text should be used to point out the relevant information that you want to discuss about the figures and to show the parts of the results/figures that back up your claims. Much of the results section is simply stating statistics that are shown in the figures.
Response 1: Many thanks to the reviewers for their perceptive and substantive comments and suggestions. A number of inappropriate figures, legends and labels in the manuscript were corrected, and the text of the manuscript was checked and revised. And we also carry out manuscript English language touch-ups prior to submission. Finally, we have carefully revised and added to the comments and suggestions made by the reviewers. (in red)
Point 2: Authors must explain what PD98059 is
Response 2: PD98059 is a specialized inhibitor of MAPK kinase. We explain this in lines 429 of the manuscript. (in red)
Point 3: Fig 4c – Ka/Ks analysis all ended up below 1. If you want to say anything, you need to determine the significance in your specific case but at face value, <1 means that these are under stabilizing selection and not under pressure to change/evolve.
The Authors state (line 150): “It is suggested that the C2H2 ZFP gene is highly conserved and consistent in the process of evolving” I’m interpreting this as the authors saying that the genes are under positive selections (in the process of evolving). I don’t think this is true based on Ka/Ks ratio. Please clarify this statement.
Response 3: We thank the reviewers for their valuable comments and we have amended this sentence to read: (line 150) ‘This suggests that the C2H2 ZFP gene is highly conserved and under stabilizing selection.’ (in red)
Point 4: Section 2.5 text:
The authors should include a description of the cultivars and the logic behind the treatments done and tissues sampled. For instance, the Tianyou cultivar does not appear anywhere else in the manuscript and it is not clear what the purpose of this variety is.
Response 4: We have added a description of the 'Tianyou-2' cultivar to line 117 of the manuscript, as well as the processing done and the logic behind the tissue sampling. We used expression data from the RNA-seq library (SRP211768), which was selected from the 'Longyou-7' and 'Tianyou-2' varieties. We would like to use this RNA-seq library to identify and screen some differentially expressed C2H2 ZFP gene in winter B. rapa under freezing stress. In this study we selected two varieties with a wide range of cold resistance, ‘Longyou-7’ and ‘Lenox’, and carried out a series of experiments. (in red)
Point 5: Figure 5 is an interesting way to show the data. The gene expression appears to be clustered by phylogenetic relationships between the TFs. However, unless this arrangement is used to show something useful, it does not make sense to present data in this way. Please include text explaining the significance of arranging the data by phylogeny. If not, it may be more useful to view the data clustered based on expression level across cultivars (similar to Figure 6)
Response 5: Many thanks to the reviewers for their comments. We have used two RNA-seq libraries to study the expression of 141 ZFP gene family members under cold and freezing stress. And we clustered the expression of the members using the phylogenetic relationships of the 141 ZFP gene family members. The aim was to identify and screen for differential genes that respond to both cold and freezing stress in the same subcluster to determine their sequence identity. (in red)
Point 6: The abbreviation of “CK” is used for the untreated control. The authors need to define this abbreviation in the main text before it is used.
Response 6: Many thanks to the reviewers for their careful comments. We have added this definition to the main text of the manuscript. Line 190: ‘The untreated rapeseed was used as a control (CK) for four abiotic stress treatments.’ (in red)
Point 7: Figure 6: Please label the different tissues used in the heatmaps. The authors use “Y”,”Z”,”J” and “G” in the column labels. I assume these refer to different tissues. It is not clear which tissues.
Response 7: We have added this figure note in Figure 6. Line 224: Y, Z, J, and G in the column labels represent leag, growth cone, stem, and root tissues of rapeseed. (in red)
Point 8: Please label the bar colors. I’m assuming they refer to the different cultivars, but this is not shown anywhere.
A and D “Relative Expression” of which gene?
Response 8: We have made changes to Figure 8. A legend on varieties has been added, and the figure titles of A and D have been modified. (in red)
Point 9: Figure 8A,D, text claims “The expression of BrZAT12 in ‘Longyou-7’ was significantly higher than that in ‘Lenox.’” The figure indicates that expression levels are the same. This also disagrees with previous data at least after the cold treatment.
Response 9: Many thanks to the reviewers for their comments. We treated 'Longyou-7' and 'Lenox' with these three solutions. We found significant differences in APX and SOD enzyme activities between 'Longyou-7' and 'Lenox' under low and room temperature treatments, but found that the expression of BrZAT12 was not significant under low and room temperature treatments. We speculate that these inhibitors have a stronger inhibitory effect on the highly expressed 'Longyou-7' than 'Lenox', a result that we need to follow up, and is only speculative at this point.
Point 10: Please mention (in the Materials and Methods) the number of replications done for each experiment.
Response 10: Many thanks to the reviewers for their valuable comments. We have added the number of replicates of each experiment to the Materials and Methods section of the manuscript. (in red)
Reviewer 2 Report
For the overexpression line, authors mentioned and demonstrated about their relative expression in the section 2.7 and supplemental figure 1E, respectively. In the text, it is mentioned that Z3 has highest expression, but it's missing in the figure.
For figure 7A and B, if authors have data with another overexpression line with minimal expression level, such as Z4 and Z9, that will be interesting to show. But, if they didn't perform that experiment, it's fine.
Author Response
Response to Reviewer 2 Comments
Point 1: For the overexpression line, authors mentioned and demonstrated about their relative expression in the section 2.7 and supplemental figure 1E, respectively. In the text, it is mentioned that Z3 has highest expression, but it's missing in the figure.
Response 1: Many thanks to the reviewers for their keen insight. We are very sorry that the correction was not made in time in the manuscript due to the numbering. We randomly selected six transgenic plants for real-time quantitative PCR and decided to use the PCR numbering of the original makes at the time of figure making. The correction has now been made in the manuscript and it is the Z5 plant that has the highest relative expression. (in blue)
Point 2: For figure 7A and B, if authors have data with another overexpression line with minimal expression level, such as Z4 and Z9, that will be interesting to show. But, if they didn't perform that experiment, it's fine.
Response 2: Comments and suggestions from the reviewers are greatly appreciated. Due to time constraints we did not phenotype the Z4 and Z9 transgenic plants. We will carry out these experiments in the next study based on the reviewers' comments to make our research results more convincing. (in blue)